# Validity Identification and Rectification of Water Surface Fast Fourier Transform-Based Space-Time Image Velocimetry (FFT-STIV) Results

**DOI:** 10.3390/s25010257

**Published:** 2025-01-05

**Authors:** Zhen Zhang, Lin Chen, Zhang Yuan, Ling Gao

**Affiliations:** 1College of Information Science and Engineering, Hohai University, Changzhou 213200, China; 221307020001@hhu.edu.cn (L.C.); yz696458@163.com (Z.Y.); 221307020030@hhu.edu.cn (L.G.); 2Key Laboratory of Hydrologic-Cycle and Hydrodynamic-System of Ministry of Water Resources, Nanjing 210024, China

**Keywords:** river surface velocity, STIV, FFT, validity identification, velocity rectification

## Abstract

Fast Fourier Transform-based Space-Time Image Velocimetry (FFT-STIV) has gained considerable attention due to its accuracy and efficiency. However, issues such as false detection of MOT and blind areas lead to significant errors in complex environments. This paper analyzes the causes of FFT-STIV gross errors and then proposes a method for validity identification and rectification of FFT-STIV results. Three evaluation indicators—symmetry, SNR, and spectral width—are introduced to filter out invalid results. Thresholds for these indicators are established based on diverse and complex datasets, enabling the elimination of all erroneous velocities while retaining 99.83% of valid velocities. The valid velocities are then combined with the distribution law of section velocity to fit the velocity curve, rectifying invalid results and velocities in blind areas. The proposed method was tested under various water levels, weather conditions, and lighting scenarios at the Panzhihua Hydrological Station. Results demonstrate that the method effectively identifies FFT-STIV results and rectifies velocities in diverse environments, outperforming FFT-STIV and achieving a mean relative error (MRE) of less than 8.832% within 150 m. Notably, at night with numerous invalid STIs at a distance, the proposed method yields an MRE of 4.383% after rectification, outperforming manual labeling.

## 1. Introduction

With global climate and environmental change, water security is becoming increasingly critical. Frequent flood disasters caused by extreme weather and topography, such as flash and basin-wide floods, have become major factors affecting economic and social stability [1]. River flow velocity is one of the key indicators in hydrological monitoring, providing significant reference value for flood disaster prevention and playing an essential role in determining other hydrological information, such as river discharge [2]. During high flood periods, rapid flow velocity, high sediment concentration, and numerous floating objects make it challenging to use traditional methods, such as rotating-element current meters, Acoustic Doppler Current Profilers (ADCPs), and the float method [3].

With the development of technology, image-based [4] and radar-based [5] river surface velocity measurement methods have gained significant attention and application. Among them, image-based methods capture videos of the river surface using cameras, calculate the motion vectors of flow tracers in the sequence of images, and convert these into the world coordinate system to measure surface flow velocity. This technology enables non-contact, instantaneous, and full-field flow velocity measurement, making it a promising tool for water flow monitoring during periods of high flooding. Common image-based methods include Large-Scale Particle Image Velocimetry (LSPIV) [6,7], Particle Tracking Velocimetry (PTV) [8,9,10], Optical Tracking Velocimetry (OTV) [11,12,13], and Space-Time Image Velocimetry (STIV) [14,15,16,17,18]. PTV and OTV have high requirements for image processing, and LSPIV is limited by the spatial resolution of different regions in oblique images. The calculation of both methods takes a long time, and the parameter settings are complex. STIV has attracted great attention due to its high resolution and strong real-time performance. STIV estimates the motion vectors by setting search lines in river videos and detecting the spatiotemporal information of tracers along these lines. The key to STIV is the detection of the Main Orientation of Texture (MOT). Depending on different methods used for MOT detection, STIV has developed several variants, including Gradient Tensor (GT) [19], Two-Dimensional Autocorrelation Function (QESTA) [20], and Fast Fourier Transform (FFT) [21] methods. Although these methods have been widely applied in practical river measurements, they are often unsuitable for complex environments, such as strong lighting, night, or rainy conditions. In these challenging scenarios, the detection results of STIV may have gross errors [22].

Fujita et al. [23] evaluate local continuity by calculating the absolute divergence value to detect and rectify PIV results. Liu et al. [24] provide parameter support for LSPIV using the float method to rectify calculation errors. However, in the field of STIV, there has not been in-depth research on methods for identifying and rectifying the validity of results. Fujita et al. [20] reduced the impact of local errors on GT results by calculating coherence values in local windows and assigning weights to each window’s results. But this approach requires different local window sizes for textures of varying lengths and noise levels. Zhang et al. [25] optimized the FFT method by adjusting parameters such as radius, direction, and bandwidth of the frequency domain filter to reduce the influence of complex noise, but the filter requires an approximate initial direction. These existing methods primarily focus on reducing the impact of noise on detection and are often limited by specific scenarios, lacking the ability to identify the validity of the results. Accurate velocity detection is critical for flow estimation and disaster forecasting. Therefore, a method is essential to evaluate and rectify velocity results.

This study analyzes the causes of gross errors in FFT-STIV results, including missing valid texture, inappropriate integration radius, and dispersion of amplitude spectrum energy. Based on the characteristics of the energy-angle distribution, three evaluation indicators—symmetry, Signal-to-Noise Ratio (SNR), and spectral width—are established. Detection results that satisfy all these indicators are considered valid velocities; otherwise, they are deemed invalid. Thresholds for SNR and spectral width are determined via statistical analysis of various datasets, enabling the method to completely filter out invalid velocities while retaining as many valid velocities as possible. The distribution law of section velocity, based on exponential distribution [26], is then used to fit the velocity curve with valid velocities, rectifying invalid results and velocities in blind areas, providing a complete velocity field. The robustness and accuracy of the proposed method are validated by experiments conducted under various scenarios at the Panzhihua Hydrological Station.

## 2. Materials and Methods

### 2.1. FFT-STIV

The time-averaged flow velocity field of a natural river surface is composed of a grid of motion vectors. The camera is set up perpendicular to the direction of the water flow to obtain video of flow movement. As shown in Figure 1a, a set of search lines is set along the direction of water flow in the image sequence. The time interval of the image sequence is Δt, with a total duration of T and M frames. Figure 1b shows the collection process of one single search line. The width and length of each search line are set to a single pixel and L pixels, respectively. A Cartesian coordinate system is established using x−t, and values of pixels along the search line are collected frame by frame to synthesize a Space-Time Image (STI) of size L×M. The movement of flow tracers within a short time is continuous, which results in directional texture features in the STI. As shown in Figure 1c, the STI exhibits prominent texture features in the form of light and dark streaks aligned in a specific direction.

The angle δ between the main trend of the texture features and the vertical axis is defined as the MOT for the STI. This angle reflects the magnitude of the time-averaged optical flow vector along the search line. Assuming that a flow tracer moves a distance of D along the direction of the search line over a duration T, this movement is mapped to d pixels in τ frames in the image coordinate system. The one-dimensional time-averaged velocity $V\left(m/s\right)$ along the search line can be expressed as:(1)$$V=\frac{D}{T}=\frac{d\cdot \Delta S}{\tau \cdot \Delta t}=\tan\delta \cdot \frac{\Delta S}{\Delta t}=v\cdot \Delta S$$
where $v\;\left(\mathrm{pixel}/s\right)$ is the optical flow vector, the sign of which indicates the direction of movement. There is a scaling relationship between V and v, determined solely by the physical-to-image scale factor $\Delta S\;\left(m/\mathrm{pixel}\right)$ of the search line, which can be obtained through flow field calibration [27]. Specifically, an imaging model of variable-height plane under oblique angle is built by introducing the tilt of camera and variation in water level. Then, a points-distance transformation between the image plane and the river surface is derived. The optical axis is required to be aligned with the cross-section when the camera is Installed, so that the azimuth angle Is set to zero. The roll and pitch angles can be calculated based on the cross-section waterline orientation method [28] or obtained through the embedded tilt sensor [29]. This method of obtaining $\Delta S\;\left(m/\mathrm{pixel}\right)$ can achieve image-free surface flow velocity measurement.

Due to the self-registration property of the Fourier transform [30], the spectral energy of a texture image, after applying the Fourier transform, is distributed along a line passing through the spectral center and orthogonal to the MOT in the spatial domain. “Self-registration” implies that the texture information formed by different targets in various directions in the spatial domain is regularized and redistributed in the frequency domain, based solely on factors such as the orientation and density of the texture units, independent of their spatial position. Based on this principle, the Fast Fourier Transform -based Space-Time Image Velocimetry (FFT-STIV) converts the problem of detecting the MOT in the spatial domain into one of finding the main orientation of the spectrum (MOS) in the frequency domain. This transformation effectively simplifies complex convolution or gradient operations in the spatial domain. The procedure is illustrated in Figure 2. By applying FFT to the STI, the amplitude spectrum image (ASI) is obtained. The FFT of a function $f\left(x,y\right)$ with dimensions L×M in Cartesian coordinates is expressed as:(2)$$F\left(u,v\right)=\sum_{x=0}^{L-1}\sum_{y=0}^{M-1}f\left(x,y\right)e^{-j2\pi \left(ux/L+vy/M\right)}=R\left(u,v\right)+jI\left(u,v\right)$$

The corresponding amplitude spectrum can be expressed as:(3)$$\left|F\left(u,v\right)\right|={\left[R^{2}\left(u,v\right)+I^{2}\left(u,v\right)\right]}^{1/2}$$

For STIs with unequal length and width, applying FFT directly will lead to “compression” of the shorter side, resulting in a directional shift in the spectrum. To address this issue, zero-padding is applied to the STI along the longer side prior to performing the FFT operation, resulting in a square STI. Assume that the size of the STI is N×N. Using the center of the ASI as the origin, and setting *R* as the integration radius, a polar coordinate system is established over the interval [0°, 180°] with an angular increment of Δθ=0.1°. The energy-angle distribution is then calculated using Equation (4). Due to the differing signal strengths across different STIs, the vertical axis in Figure 2d represents normalized values. For the same ASI, different *R* values will result in different energy-angle distributions, and the accumulated energy P(θ) may vary as well. ASIs with long central spectral lines and concentrated energy require less stringent selection of *R*. However, for ASIs with dispersed energy or shorter spectral lines, *R* needs to be carefully adjusted; otherwise, it becomes difficult to distinguish between valid signals and noise.
(4)$$P(\theta )=2\sum_{r}^{R}F^{2}(r,\theta )/N$$

F(r,θ) is the amplitude distribution, and P(θ) is the energy value corresponding to the angle θ. In the histogram, the angle corresponding to the peak value $P_{s}$ is defined as the MOS, which is orthogonal to the MOT. For rivers with a flow direction from left to right, the effective spectral signal lies within the interval (90°, 180°). Conversely, for flow from right to left, it lies within (0°, 90°). This study uses a river flowing from left to right as an example. The peak value is not simply defined as the “maximum energy”, but rather as the point of the maximum local extremum where energy first increases and then decreases within a given region. If the P(θ) is the extremum within a 0.5° neighborhood, it is defined as a peak, and the highest of these peaks is $P_{s}$. Typically, the MOS of STIs from natural rivers has a range between (95°, 175°) [22]. To eliminate interference from the energies at 90° and 180°, the search range is limited accordingly. Thus, the search for $P_{s}$ follows as:(5)$$\left\{\begin{array}{l} 95^{\circ }<\theta <175^{\circ }\\ P_{\theta }>\mathrm{Mean}(P_{i}),\;\theta -0.5<i<\theta +0.5,\;i\neq \theta \\ P_{s}=\mathrm{Max}(P_{\theta }) \end{array}\right.$$
where Mean is the average value of the set, and Max is the maximum value in the set.

### 2.2. Analysis of False Detection

#### 2.2.1. Missing Valid Texture

According to Section 2.1, the movement of flow tracers presents oblique textures on the STI, while stationary objects present vertical textures. Therefore, the oblique textures formed by the movement of tracers with consistent angles are called valid texture, and the corresponding STIs are valid STIs. During insufficient lighting (e.g., nighttime), extreme weather (e.g., rainy) and obstructions may result in the absence of valid textures in the STI, leading to no valid bright spectral lines in the corresponding ASI. Consequently, FFT-STIV may yield erroneous detection results, such as shown in Figure 3. The black line in Figure 3b represents the detection result of FFT-STIV corresponding to the peak in Figure 3c.

#### 2.2.2. Inappropriate Integration Radius

The key to FFT-STIV lies in determining the integration radius, as it defines the range within the ASI in which signals are searched. Currently, there is no effective method for adaptively selecting the integration radius, resulting in the MOT detection in complex scenes during actual flow measurements being constrained by a fixed integration radius. An integration radius of N/2 is suitable for most cases when the central spectral line in the ASI is long and bright. However, in cases of weak effective signals, such as insufficient lighting at nighttime or overly strong light on sunny days, the valid textures become less distinct. Additionally, rainy days can disrupt the continuity of the effective texture. As a result, the central spectral line becomes shorter, making it difficult to distinguish valid signals from noise, especially when using a large integration radius. Figure 4 shows the detection process in a rainy case. The black lines in ASIs are the FFT-STIV results corresponding to the Peak in (d) and (e). Since the central spectral line is short and the energy is dispersed, *R* = N/2 will introduce more noise signals, resulting in a lower signal-to-noise ratio. *R* = N/4 reduces the collection of noise signals, so the correct Peak can be detected. For good conditions, effective detection can be performed using *R* = N/2 or *R* = N/4, as shown in Figure 5.

#### 2.2.3. The Dispersion of ASI Energy

The flow pattern of natural rivers is highly complex and variable, and the consistency of the textures presented in the STI greatly affects the central spectral line in the ASI. When the texture consistency of the STI is poor, the energy distribution in ASI is dispersed. As shown in Figure 6, in such cases, the angle of the valid signal in the ASI spans a range, and no unique solution exists.

### 2.3. Velocity Rectification Method

#### 2.3.1. Evaluation Indicators

The values in the energy-angle distribution are obtained by summing the pixel values in each direction of the ASI, thereby incorporating both valid texture signals (i.e., the bright central spectral line) and complex background noise. The valid signal is characterized by a peak in the histogram, based on which three evaluation indicators are established—symmetry, SNR, and spectral width—to identify false detections as described in Section 2.2.

(1)Symmetry

In FFT-STIV, the valid signal is characterized by the energy value $P_{s}$ as the detected peak. Reference [25] analyzes the energy-angle distribution characteristics of STIs in various scenarios. When there is an effective peak, the energy near the peak presents an approximate one-dimensional Gaussian distribution; that is, it has an approximately symmetric characteristic. Since the energy in ASI is mainly concentrated in the vertical, horizontal, and valid signal, the energy changes less between the peak and the closer boundary; there is a “valley” between them and $P_{s}$. The minimum valley value, $P_{n}$, which lies between the detected peak and the closer boundary of the interval, is defined as the background noise, and Figure 7a,b provide two examples. There should be a point on the other side that has the same energy value as $P_{n}$, which is defined as the symmetric point of the valley. Based on this concept, the symmetry indicator is established. The angle difference between the peak and the valley is defined as B, and the range of searching for symmetric points on the other side of the peak is limited to 2B to satisfy a one-dimensional Gaussian shape. As shown in Figure 7c, if the symmetric point does not exist, it indicates that a valid peak is not present in the histogram, and the symmetry indicator is deemed invalid.

The search for $P_{n}$ is conducted within the specified interval using a similar approach as that for searching $P_{s}$, as shown by the following equation:(6)$$\left\{\begin{array}{l} \theta_{l}<\theta <\theta_{r}\\ P_{\theta }<\mathrm{Mean}(P_{i}),\;\theta -0.5<i<\theta +0.5,\;i\neq \theta \\ P_{n}=\mathrm{Min}(P_{\theta }) \end{array}\right.$$
where $\theta_{l}$ and $\theta_{r}$ represent the left and right boundaries, respectively, which could be either the angle corresponding to the peak or the boundary of the interval for searching the MOS.

(2)SNR

SNR is an important metric in signal analysis, commonly used to measure the prominence of the valid signal against background noise, specifically represented as the ratio of valid signal to noise. In this study, the energy ratio of the peak and valley is defined as SNR, which is calculated as shown in Equation (7). The SNR of valid and invalid STIs is statistically analyzed, and a threshold Th1 is set to filter out low-SNR data such as spurious spikes or gentle signal, while retaining as many valid STIs as possible. When SNR≤Th1, the indicator is marked as invalid.
(7)$$SNR=\frac{P_{s}}{P_{n}}$$

(3)Spectral width

In complex noisy environments, the energy-angle distribution curve may not be smooth, and spurious spikes can create false valley-peak-valley patterns, as shown in Figure 7d. To address this issue, the spectral width indicator is proposed, which is defined as the absolute difference in angle between the two valleys. Under valid conditions, the spectral width is relatively large, whereas the spectral width formed by spurious spikes is much smaller. The spectral width of valid and invalid STIs is statistically analyzed, and a threshold Th2 is set. When spectral width≤Th2, the indicator is marked as invalid.

#### 2.3.2. Distribution Law of Section Velocity

Since all measurement points are located along the same water level, their velocities follow a certain pattern. The distribution law of section velocity, based on the exponential distribution, is chosen [26]. As shown in Figure 8, the section is divided into two parts using the point of maximum velocity as the boundary. On one side of the section, the velocity at each measurement point follows an exponential relationship with its distance from the nearest riverbank. Common sections are often rectangular, U-shaped, or trapezoidal in profile, so the maximum velocity typically occurs near the center. The velocity relationship is therefore simplified as follows:(8)$$v(x)=k{\left(\frac{x-x_{0}}{W/2}\right)}^{z}$$
where x is the distance between the measurement point and the reference point and $x_{0}$ is the distance between the closer bank and the reference point. The reference point is a fixed point artificially set on one side of the bank. W is the width of the water surface between the two banks, and v(x) is the flow velocity at the measurement point. k and z are unknown parameters, k reflects the scaling relationship with the maximum velocity, and z reflects the exponential distribution characteristics of the section velocity. Taking the left side as an example, a set of valid data $\left\{x_{1},x_{2},\cdots ,x_{n}|x_{l}\leq x_{n}\leq x_{mid}\right\}$, $\left\{v(x_{1}),v(x_{2}),\cdots ,v(x_{n})|x_{l}\leq x_{n}\leq x_{mid}\right\}$ is obtained based on the three evaluation indicators outlined in Section 2.3.1. Then, k and z are calculated by the Least Squares Method as follows:(9)$$\left\{\begin{array}{l} X^{T}X=\left[\begin{array}{c} n\\ \sum_{i=1}^{n}\ln(\frac{x_{i}}{W/2}) \end{array}\begin{array}{c} \sum_{i=1}^{n}\ln(\frac{x_{i}}{W/2})\\ \sum_{i=1}^{n}{\left(\ln(\frac{x_{i}}{W/2})\right)}^{2} \end{array}\right]\\ X^{T}y=\left[\begin{array}{c} \sum_{i=1}^{n}\ln v(x_{i})\\ \sum_{i=1}^{n}\ln(\frac{x_{i}}{W/2})\ln v(x_{i}) \end{array}\right]\\ \beta =\left[\begin{array}{c} \ln k\\ z \end{array}\right]\\ X^{T}X\beta =X^{T}y \end{array}\right.$$

The velocity curve fitting process on the right side is similar to the case on the left side above. The undetermined velocities are then calculated based on the fitted curve. The velocities corresponding to STIs deemed invalid based on the evaluation indicators, or those in blind areas, can be calculated using the fitted velocity curves.

## 3. Results

### 3.1. Evaluation Criteria

Using current meter data as a reference, the proposed method, FFT-STIV, and the velocities converted from manually labeled MOT are compared. The evaluation is based on the following three criteria.

Mean Absolute Error (MAE) reflects the average magnitude of errors. It is calculated as follows:(10)$$\mathrm{MAE}=\frac{1}{n}\sum_{i=1}^{n}\left|y_{i}-{\hat{y}}_{i}\right|$$

Root Mean Squared Error (RMSE) is the arithmetic square root of the Mean Squared Error (MSE). It is sensitive to large errors, making it effective for identifying anomalies in the detection results. The equation for calculating RMSE is as follows:(11)$$\mathrm{RMSE}=\sqrt{\frac{1}{n}\sum_{i=1}^{n}(y_{i}-{\hat{y}}_{i}{)}^{2}}$$

Mean Relative Error (MRE) is the average of relative errors, indicating the proportion of the error compared to the actual value. It is calculated using the following equation:(12)$$\mathrm{MRE}=\frac{1}{n}\sum_{i=1}^{n}\left|\frac{y_{i}-{\hat{y}}_{i}}{y_{i}}\right|$$
where $y_{i}$ is the measurement of current meter, ${\hat{y}}_{i}$ is the measurement of STIV, n is the total number of observations, and i is the serial number of search lines.

### 3.2. Overview of Gauging Site

As shown in Figure 9, the Panzhihua Hydrological Station, located in Panzhihua City, Sichuan Province, was selected as the experimental gauging site. It is an important hydrological and reporting station for the Jinsha River which is the main stream of the Yangtze River, equipped with a hydrological cableway and comprehensive flow measurement facilities. The section monitored by the hydrological station is a relatively stable “U”-shaped profile, with both banks composed of boulders. The riverbed is characterized by a large amount of gravel, which is typical of a mountain riverbed. Unlike the gentle silt-laden rivers of the plains, the water here flows turbulently, crashing against rough rocks and forming waves and swirling patterns. These natural flow tracers provide ideal conditions for image-based flow measurement.

The online measurement system is set up on the slope on the side of the hydrological station as shown in Figure 10a, which is located between the cableway section and the staff gauge section. The camera’s tilt angle is set to 19.8° to ensure adequate field of view. Remote communication is achieved through 4G network and VPN, while system control and video data storage and processing are carried out remotely using specialized software. Figure 10b shows the section captured by the camera (Hikvision, DS-2CD3T86FWDV2-I3S, 4 mm), with a resolution of 3840 × 2160 (px2), and river flow moving from left to right relative to the camera. Each measurement video lasts 30 s, with a frame rate of 25 fps. Search lines are set at intervals of 1 m, and the length of each search line in the image sequence is set to 750 pixels to generate STIs of size 750 × 750 (px2). The water level in the experiments ranged from 986.09 m to 999.74 m, with the minimum ΔS near the bank being 0.005 (m/pixel) and the maximum ΔS at the far bank being 0.065 (m/pixel).

### 3.3. Statistica Analysis of Evaluation Indicators

A total of 47 river surface videos were randomly collected under varying water levels, weather, and lighting conditions, resulting in 6758 STIs, of which 2142 STIs were deemed invalid. All valid STIs satisfy the symmetry indicator, which was therefore considered as a necessary condition. In the invalid STIs, the screening rate by symmetry indicator was 23.16%, indicating that 496 STIs did not satisfy the indicator. To determine Th1 and Th2, a statistical analysis was performed on the SNR and spectral width of 1646 invalid STIs that satisfy the symmetry indicator. The results are shown in Figure 11, the vertical axis represents the proportion of STIs within the corresponding SNR or spectral width range. The SNR ranges from [1.0019, 1.2350], while the spectral width ranges from [0.1467, 8.4315].

To avoid invalid STIs being misclassified as valid, which could lead to erroneous velocity rectification, the core principle for selecting Th1 and Th2 is to eliminate invalid STIs while retaining as many valid STIs as possible. Based on the analysis shown in Figure 11, the thresholds were chosen as Th1=1.15 and $Th2=3.5^{\circ }$. The screening rates and false detection rates based on the symmetry indicator are summarized in Table 1. The results indicate that setting the thresholds to higher values (e.g., the maximum statistical values) effectively filters out invalid STIs but misclassifies more valid STIs. The screening rate of a single indicator does not reach 100% with the selected Th1 and Th2. However, the combined use of both indicators successfully eliminates all invalid STIs while retaining 99.83% of the valid STIs.

### 3.4. Velocity Comparison Experiments

Comparative experiments were conducted under three different scenarios: rainy, sunny, and nighttime, to verify the accuracy of the proposed method. To handle varying forms of the central spectral line in the ASI and reduce the false detection rate of FFT-STIV across different scenarios, N/4 was used as the integration radius. The rotating-element current meter (LS25-3A) was used to measure flow velocities as reference. The experiment used the vertical average velocity obtained by each method to compare. The current meter data comes from the hydrological station records. The velocities of fixed measurement points at the relative water depth of 0.2 and 0.8 are measured, respectively, using hydrological cableway, and the average velocity is obtained using the two-point method. The results obtained by STIV are the surface velocities, which are multiplied by the water surface velocity coefficient as the final result for comparison. The water surface velocity coefficient at the corresponding water level is the ratio of the measured flow rate to the virtual flow rate obtained by the surface velocity.

#### 3.4.1. Rainy

Test 1: The flow measurement was conducted at 9:00 a.m. on 19 August 2020, during heavy rain, with the river at a water level of 999.2 m. Figure 12a shows the view of section and the arrangement of the search lines, while Figure 12b presents the velocity using different methods at each measurement points. “Distance” represents the distance from the reference point. “FFT-STIV” is obtained by Section 2.1, “ours” is the results after identification and rectification based on “FFT-STIV”, and “manual” is calculated by the STIV principle after manually marking MOT. For positions between 30 m and 165 m, the STIs contain textures that can be manually labeled, while the remaining STIs lack textures due to occlusions, turbulence, and focal length issues. The proposed method identifies STIs beyond 155 m as invalid and subsequently rectifies the flow velocity, resulting in measurements closer to the current meter data compared to FFT-STIV. However, the results from “manual”, “FFT”, and “ours” all show a sharp decline starting at 155 m, which follows the same trend as the current meter data but with a significant error. This is due to the gross error in the scale factor beyond 150 m, which can reach up to 10% [31]. Figure 13a and Figure 13b, respectively, show the section velocity of FFT-STIV and the proposed method. FFT-STIV gives extremely low velocity for STIs with missing textures, while the proposed method accurately identifies the validity of results and applies corrections. Notably, the proposed method also provided velocity estimates for the near-bank blind area shown in Figure 12a.

Table 2 presents the error analysis of each method using the current meter data as a reference, where “manual” excludes STIs above 165 m due to the inability to label them. For the range of 30 m to 150 m, where FFT-STIV results are valid, the proposed method maintains consistent results with FFT-STIV. However, in the distant region, due to factors such as cloudy weather and scale factor inaccuracies, the FFT-STIV error is relatively large, while the proposed method shows improvements in MAE, RMSE, and MRE.

#### 3.4.2. Sunny

Test 2: The flow measurement was conducted at 2:18 p.m. on 15 April 2021, under sunny conditions, with the river at a water level of 986.35 m and the surface largely covered by tree shade. The STIs from 50 m to 160 m contained textures that could be manually labeled, while the remaining STIs lacked textures due to heavy shading. As shown in Figure 14b, “FFT” only accurately detected velocities at 90 m, 105 m, and 120 m. Since reflections and shadows reduce texture clarity and degrade SNR, FFT-STIV with a fixed integration radius struggled to accurately detect STIs, while the proposed method was able to identify these regions and provide rectified results. Moreover, “manual” showed gross errors due to scale factor inaccuracies and random error, particularly for the far-bank region. In contrast, the rectified results were closer to the current meter data. As shown in Figure 15, the proposed method gives favorable section velocity based on the distribution law of section velocity. This is particularly evident for the far-bank region, where a small number of accurate velocity values provided an effective reference for curve fitting.

Due to extreme environmental factors such as heavy shadows, the MRE of FFT-STIV exceeded 50%. In contrast, the proposed method maintained an MRE of 9.396% even for the entire section velocity, as shown in Table 3.

#### 3.4.3. Nighttime

Test 3: The flow measurement was conducted at 8:00 p.m. on 19 August 2020, with the river at a water level of 999.74 m, and illumination provided by a supplementary light from the opposite bank. The STIs from 30 m to 175 m contained textures that could be manually labeled. As shown in Figure 16b, poor lighting conditions beyond 150 m caused blurred textures in the STIs, resulting in short and dispersed spectral lines, making FFT-STIV fail. Due to inaccuracies in the scale factor, the results for “manual” at 165 m and 175 m had larger errors compared to “ours”. As shown in Figure 17, FFT-STIV only performed well in areas with adequate lighting and was unable to provide a complete section velocity field for flow estimation.

Table 4 indicates that the MRE of the proposed method was only 4.383%, outperforming both FFT-STIV and the manual values. Additionally, the results also surpassed those of Test 1, primarily because the reliable velocities near the central flow path were used for curve fitting, thereby avoiding scale factor errors in the distant region.

### 3.5. Method Tests in Various Scenario

To verify the robustness of the proposed method, additional tests were conducted under varying time periods and water levels based on the three scenarios described in Section 3.4. The section velocity distributions obtained using the FFT-STIV method and the proposed method were compared. The results are presented in Table 5, where the test numbers follow the experiments in Section 3.4. For the results of each test, the upper shows the result from the FFT-STIV, and the lower shows the result from the proposed method. The proposed method estimates velocities in the near-bank blind zones in Test 1, 3, 4, 5, and 9. However, if valid velocities cannot be obtained in areas where velocity changes may occur, the proposed method may fail to provide an accurate velocity trend. For instance, near-bank velocities were overestimated in Test 2 and Test 8. In Test 5, high-speed moving shadows caused by cars on the far bank under oblique lighting created more pronounced texture signals in both STI and ASI. This led to higher velocities, which subsequently influenced the velocity rectification process.

In general, the proposed method effectively detects false detections caused by FFT-STIV failing to locate the peak and successfully rectifies erroneous velocities. However, reflections caused by dynamic objects on the bank can interfere with texture detection, which can be mitigated by introducing prior information, such as the maximum velocity, as a constraint. Additionally, the accuracy of the MOT detection method still needs to be improved to obtain more precise interpolated flow velocities.

## 4. Conclusions

This paper proposes a method for validity identification and rectification of FFT-STIV results. Based on the energy-angle distribution curve generated by FFT-STIV, three evaluation indicators were established: symmetry, SNR, and spectral width. Only results that satisfy all indicators are considered valid data. All valid results satisfy the symmetry indicator, and the thresholds of SNR and spectral width are obtained through statistical analysis of various STIs. Finally, the SNR threshold Th1=1.15 and the spectral width threshold $Th2=3.5^{\circ }$ were established. The three indicators can eliminate all invalid velocities and retain 99.83% of the valid velocities. Using a larger Th1 or Th2 can make each individual indicator more effective at eliminating invalid velocities, but it may also result in more valid velocities being incorrectly classified as invalid.

The distribution law of section velocity based on the exponential distribution is introduced, using valid velocities and corresponding positions to fit the velocity curve, thereby rectifying invalid velocities and estimating velocities in blind areas. The proposed method was tested at the Panzhihua Hydrological Station under different water levels, weather conditions, and lighting scenarios. When FFT-STIV performed well, the proposed method helped reduce errors caused by undetectable STIs. Under extreme conditions, such as strong lighting or low water levels, it effectively identified false detection in FFT-STIV results. In nighttime conditions with insufficient lighting, the method provided corrections, resulting in more accurate outcomes than those obtained from manual labeling, with an MRE of just 4.383%. It is worth noting that experiments have shown that, when the scale factor error is large, the results given by the proposed method are better than those converted by the right MOT.

However, the selection of Th1 and Th2 in the proposed method depends on the analysis of existing data, and different rivers may require different thresholds. The lateral velocity rectification method selected in this paper is a simplified one, which is applicable to sections that are rectangular, U-shaped, or trapezoidal in profile with maximum velocity near midstream. For other sections, it is necessary to search for the location of the maximum velocity and ensure the effectiveness of the velocities around it. Moreover, the proposed method can only identify the validity of results but cannot further improve the detection of falsely detected STIs that contain textures. Future work will focus on developing optimized detection methods for such falsely detected STIs, as well as refining the selection of valid data to achieve better velocity curve fitting, particularly in scenarios involving significant scale factor errors.

## Figures and Tables

**Figure 1 sensors-25-00257-f001:**
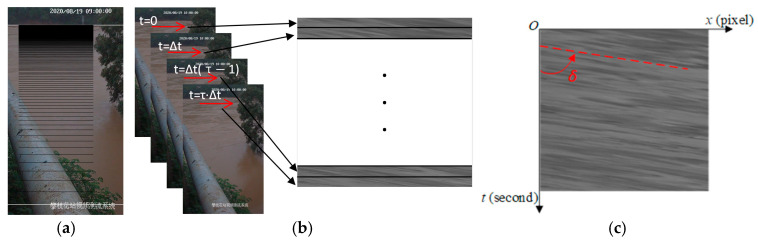
Process of STI synthesis: (**a**) search lines; (**b**) image sequences; (**c**) STI.

**Figure 2 sensors-25-00257-f002:**
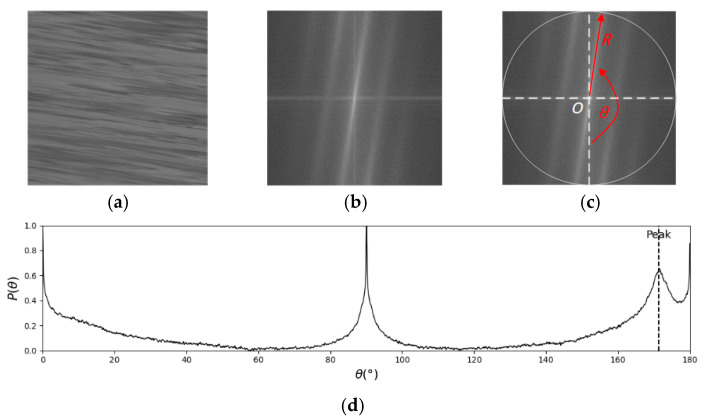
Diagram of FFT method: (**a**) STI; (**b**) ASI; (**c**) polar coordinate system; (**d**) histogram of energy-angle distribution.

**Figure 3 sensors-25-00257-f003:**
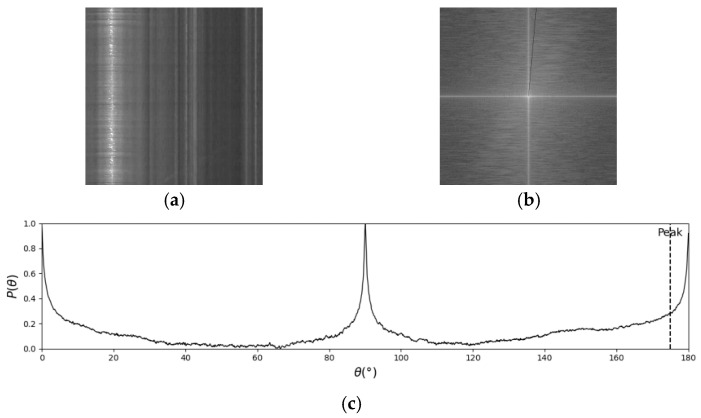
False detection caused by missing texture: (**a**) STI; (**b**) ASI; (**c**) histogram of energy-angle distribution.

**Figure 4 sensors-25-00257-f004:**
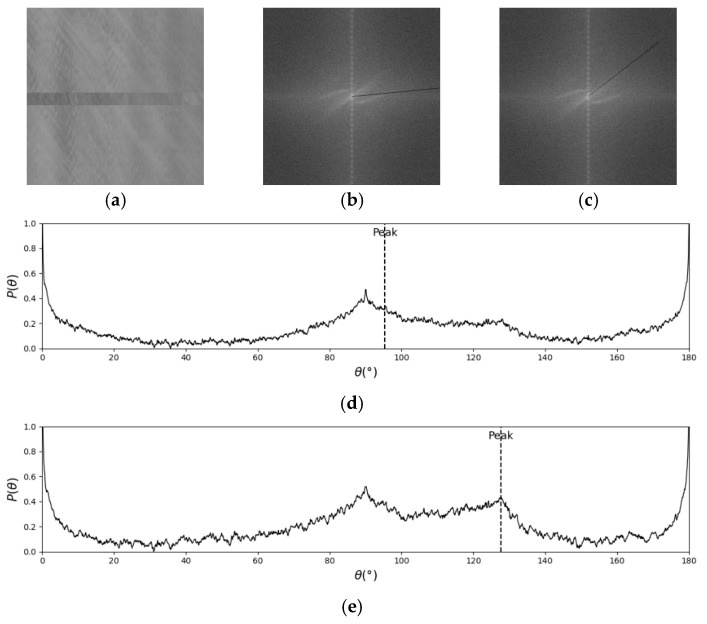
False detection caused by inappropriate integration radius: (**a**) STI; (**b**) ASI with R = N/2; (**c**) ASI with R = N/4; (**d**) histogram of energy-angle distribution with R = N/2; (**e**) histogram of energy-angle distribution with R = N/4.

**Figure 5 sensors-25-00257-f005:**
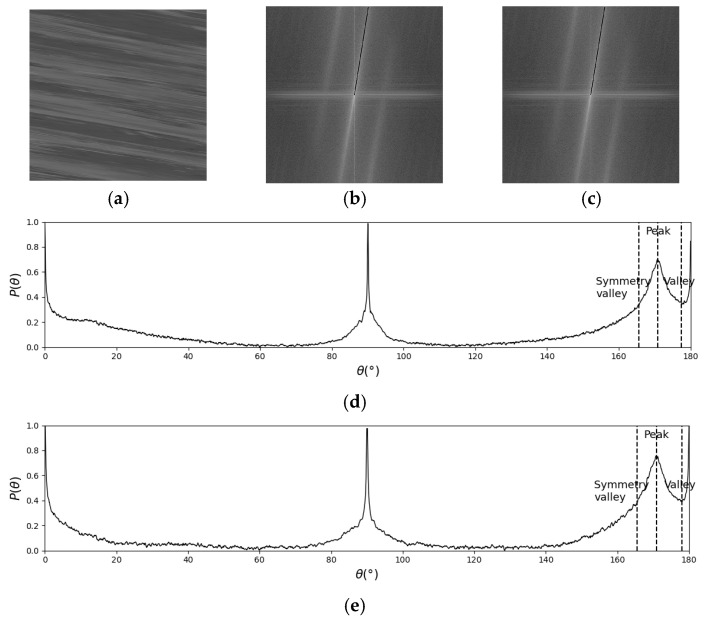
Detection results in a good condition: (**a**) STI; (**b**) ASI with R = N/2; (**c**) ASI with R = N/4; (**d**) histogram of energy-angle distribution with R = N/2; (**e**) histogram of energy-angle distribution with R = N/4.

**Figure 6 sensors-25-00257-f006:**
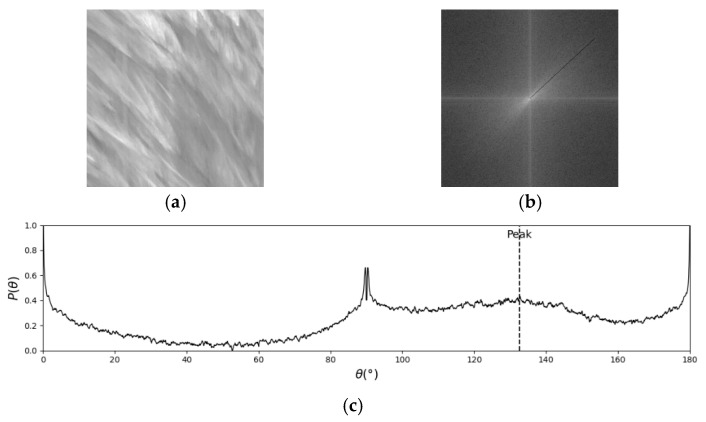
False detection caused by the dispersion of ASI energy: (**a**) STI; (**b**) ASI; (**c**) histogram of energy-angle distribution.

**Figure 7 sensors-25-00257-f007:**
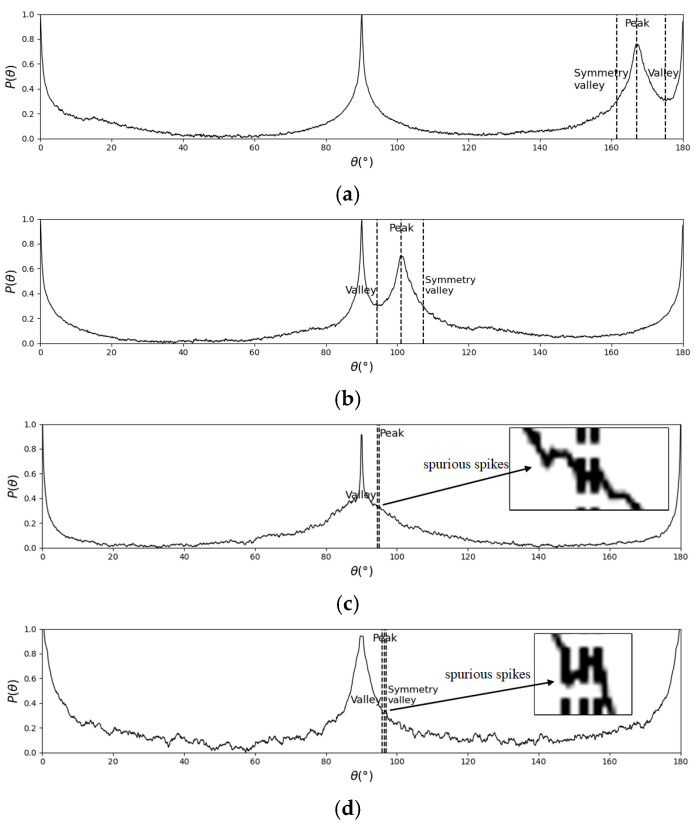
Typical histogram of energy-angle distribution: (**a**) valid result with valley close to 180°; (**b**) valid result with valley close to 90°; (**c**) the absence of symmetry valley; (**d**) false valley-peak-valley pattern caused by spurious spikes.

**Figure 8 sensors-25-00257-f008:**
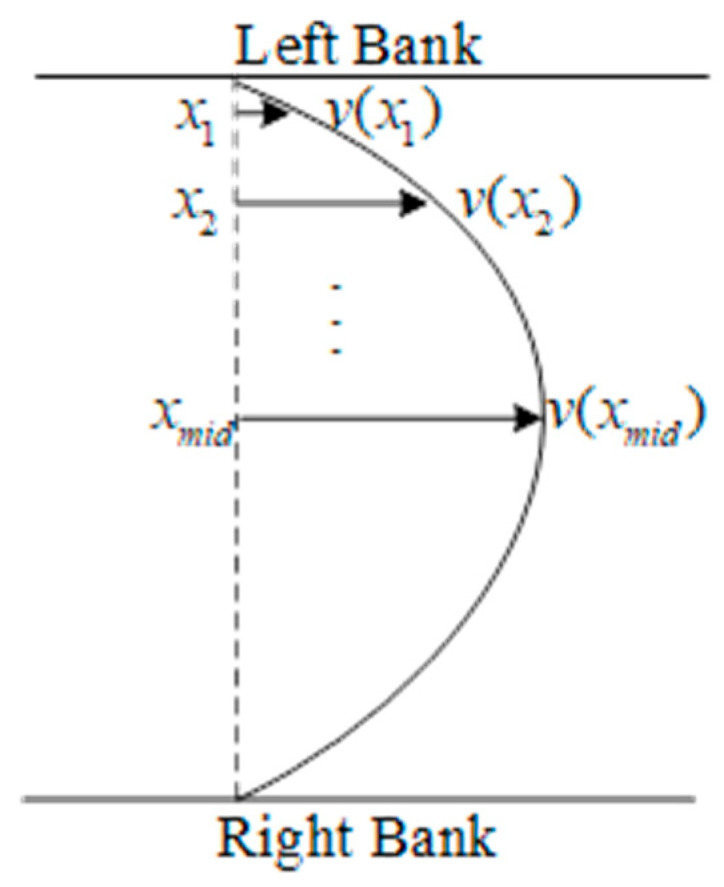
Section velocity distribution.

**Figure 9 sensors-25-00257-f009:**
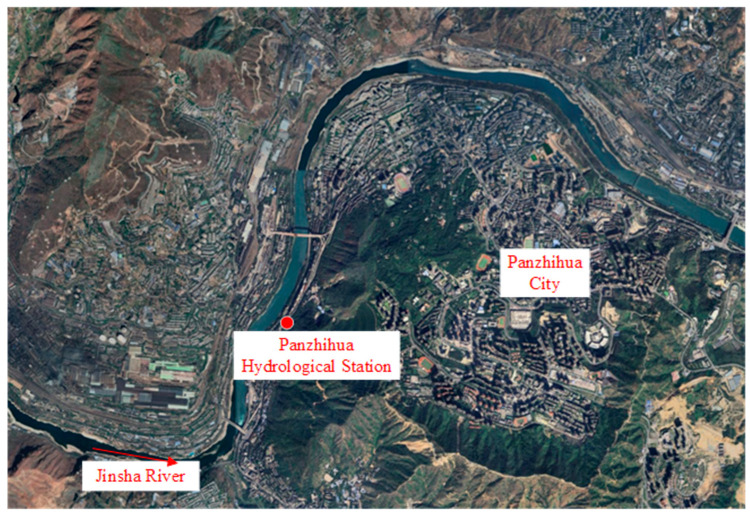
The location of gauging site.

**Figure 10 sensors-25-00257-f010:**
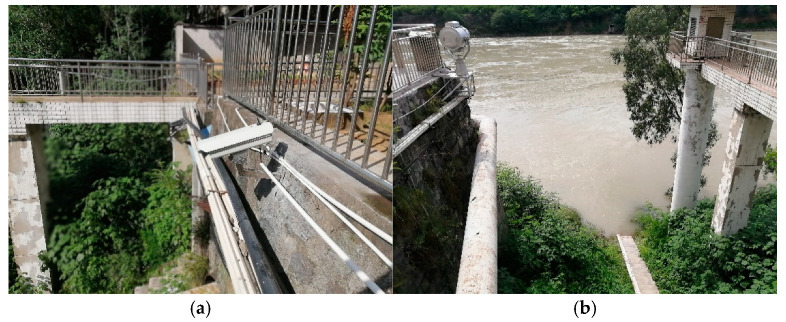
System setup and section view: (**a**) measurement system; (**b**) section view.

**Figure 11 sensors-25-00257-f011:**
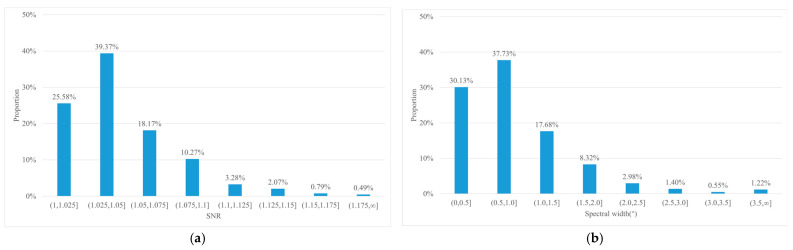
Statistical data of SNR and spectral width for invalid STIs that satisfy the symmetry indicator: (**a**) SNR; (**b**) spectral width.

**Figure 12 sensors-25-00257-f012:**
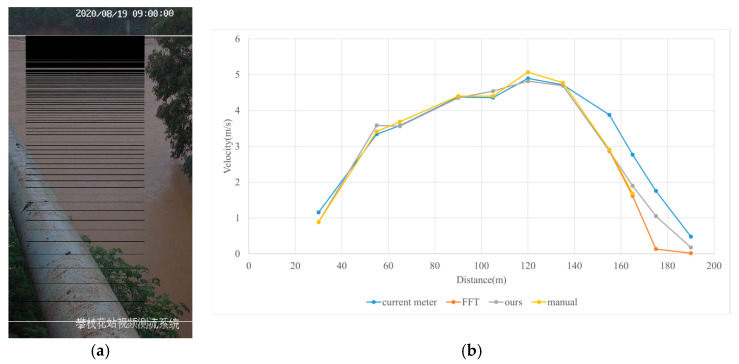
Velocity comparative experiment in Test 1: (**a**) search lines; (**b**) results of different methods.

**Figure 13 sensors-25-00257-f013:**
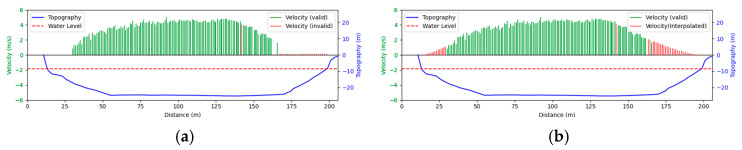
Section velocity distribution in Test 1: (**a**) FFT-STIV; (**b**) ours.

**Figure 14 sensors-25-00257-f014:**
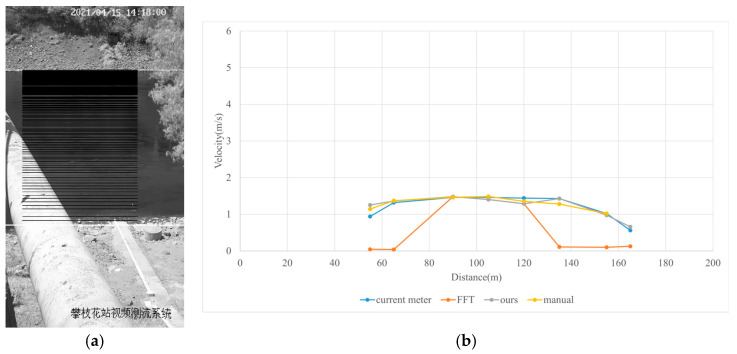
Velocity comparative experiment in Test 2: (**a**) search lines; (**b**) results of different methods.

**Figure 15 sensors-25-00257-f015:**
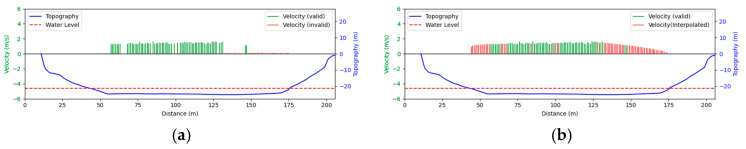
Section velocity distribution in Test 2: (**a**) FFT-STIV; (**b**) ours.

**Figure 16 sensors-25-00257-f016:**
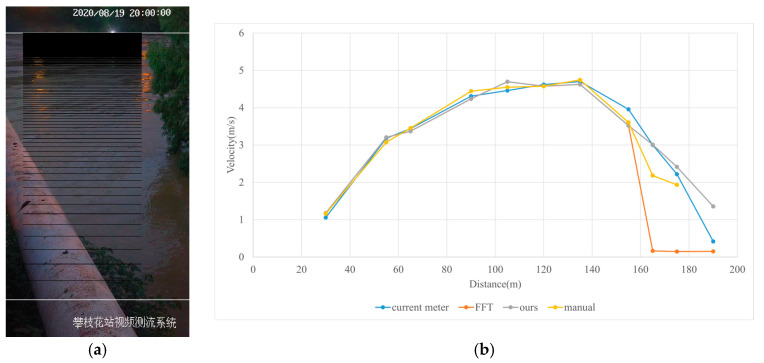
Velocity comparative experiment in Test 3: (**a**) search lines; (**b**) results of different methods.

**Figure 17 sensors-25-00257-f017:**
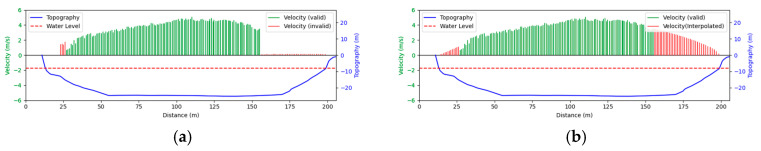
Section velocity distribution in Test 3: (**a**) FFT-STIV; (**b**) ours.

**Table 1 sensors-25-00257-t001:** Screening rates and false detection rates based on the symmetry indicator.

STIs	SNR ≤ 1.15	Spectral Width ≤ 3.5°	SNR ≤ 1.15 || Spectral Width ≤ 3.5°	SNR ≤ 1.24	Spectral Width ≤ 8.5°	SNR ≤ 1.24 || Spectral Width ≤ 8.5°
Screening rates of invalid STIs that satisfy the symmetry indicator	98.72%	98.78%	100%	100%	100%	100%
False detection rates of valid STIs	0.17%	0	0.17%	1.32%	1.71%	2.11%

**Table 2 sensors-25-00257-t002:** Error analysis in Test 1.

Method	MAE (m/s)	RMSE (m/s)	MRE (%)	MRE Within 150 m (%)
FFT	0.463	0.718	19.788	5.437
Ours	0.343	0.495	13.560	5.437
Manual	0.312	0.500	11.027	5.001

**Table 3 sensors-25-00257-t003:** Error analysis in Test 2.

Method	MAE (m/s)	RMSE (m/s)	MRE (%)	MRE Within 150 m (%)
FFT	0.636	0.810	58.652	50.151
Ours	0.092	0.132	9.396	8.832
Manual	0.090	0.112	7.851	7.322

**Table 4 sensors-25-00257-t004:** Error analysis in Test 3.

Method	MAE (m/s)	RMSE (m/s)	MRE (%)	MRE Within 150 m (%)
FFT	0.599	1.123	22.260	3.391
Ours	0.129	0.178	4.383	3.391
Manual	0.198	0.303	6.893	2.854

**Table 5 sensors-25-00257-t005:** Results of the method tests in various scenarios.

No.	Search Lines	Water Level (m)	Scenario	Results
4	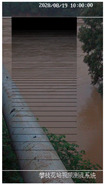	999.25	Rainy	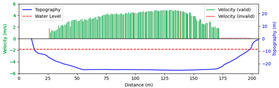 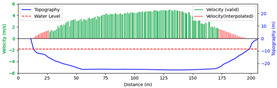
5	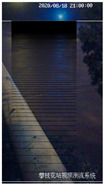	996.83	Rainy and Nighttime	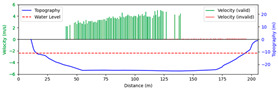 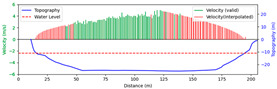
6	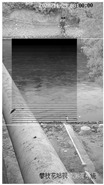	986.09	Sunny	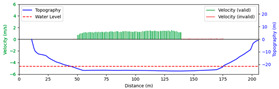 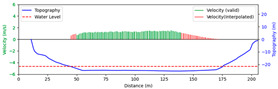
7	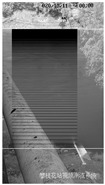	991.69	Sunny	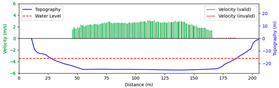 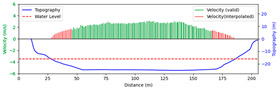
8	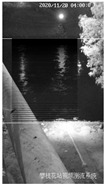	986.00	Nighttime	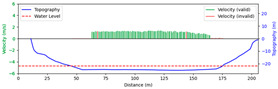 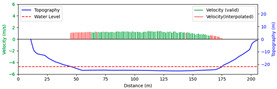
9	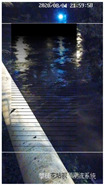	995.15	Nighttime	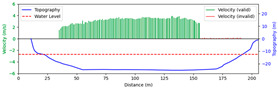 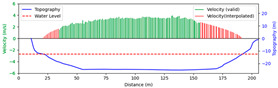

## Data Availability

The data presented in this study are available on request from the corresponding author. (The data involves topographic data and hydrological information of key hydrological stations).

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
