# Peer review of "Validity Identification and Rectification of Water Surface Fast Fourier Transform-Based Space-Time Image Velocimetry (FFT-STIV) Results"

_sensors, 2025, doi:10.3390/s25010257_

Round 1
Reviewer 1 Report
Comments and Suggestions for Authors
The study presents a novel approach that enhances the post-processing of FFT-STIV results by assessing and optimizing their validity, eliminating erroneous detections, and improving the reliability of flow velocity measurements. This approach is innovative and contributes to enhancing FFT-STIV performance. However, several issues remain that need to be addressed:
1. The abstract needs improvement. It provides excessive general background information and fails to clearly highlight the specific research problem being addressed and the key contributions made by the study.
2. The introduction needs revision to better emphasize the focus and novelty of the research. The authors should more clearly highlight the unique contributions of the study and establish its significance within the existing body of work.
3. In Section 2.2.2, the authors discuss the impact of the integration radius on the recognition of the MOT. However, this topic is not explored further in the manuscript. What is the significance of addressing the integration radius here? Additionally, in Line 174, the authors claim that reducing the integration radius under rainfall conditions improves the accuracy of MOT recognition. It would be helpful to provide experimental evidence or references to support this claim. Similarly, in Section 2.2.3, the authors mention turbulence and backflow issues but do not revisit or elaborate on them in subsequent sections. These points should either be clarified or removed if not further addressed. In the results section, the authors present three scenarios: rainy, sunny, and nighttime. Therefore, Section 2.2.2 should focus more on the challenges that persist in these specific conditions.
4. In Section 2.3, the authors propose an indicator to measure the ratio of noise to valid information to determine whether the valid information in the STI image is prominent. This approach seems reasonable. However, in Line 223, the authors state: "It is stipulated that when SNR > 1, the peak is considered prominent against the background noise, indicating that the indicator is valid; otherwise, it is deemed invalid." Since Ps will always be greater than Pn (with Ps representing the detected peak and Pn representing the minimum valley value), the SNR will always be greater than 1, rendering this indicator ineffective. The authors should provide clarification and an explanation for this issue.
5. In Line 235, the authors state, "there should be a point on the other side that has the same energy value as Pn, which is defined as the symmetric point of the valley." What is the basis for this assertion? What underlying mechanism supports this claim? Are there any statistical analyses or evidence to substantiate this statement?
6. In Line 259, the authors mention the "lack of valid texture." How the authors define and determine a lack of valid texture in the STI? A more detailed explanation of this criterion is needed.
7. The authors should provide a more detailed analysis of the effectiveness of the evaluation indicators introduced in Section 2.3.1. Specifically, how many STIs were considered invalid or uninformative? Were there any false positives or misjudgments during the process? If so, which indicators contributed to these errors?
8. The raw FFT results should be presented, as well as the results after applying the evaluation indicators. Additionally, the FFT results for the "distribution law of section velocity," as described in Section 2.3.2, also should be included for a comprehensive evaluation of the approach. Without these details, it is difficult to assess the individual effectiveness of the methods presented in Sections 2.3.1 and 2.3.2.
9. In Figures 13(a), 15(a), and 17(a), the background images are shown without projection transformation, whereas the velocity measurement lines appear to have undergone projection transformation. The coordinate systems should be unified for consistency, and the authors should redraw these figures accordingly. Furthermore, Figure 16(a) is presented as a grayscale image, while Figure 17(a) lacks flow velocity lines. Additionally, why are there so few data points in Figures 13(b), 15(b), and 17(b), and why is the spacing between points not uniform? How were the velocity measurement lines set by the authors?
10. In Lines 323-325, the authors mention basic information about STI acquisition but do not provide details about the real-world length represented by each pixel in the STI. This information is essential for accurately converting STI data to flow velocities, and the authors should include this detail.
11. In Lines 41-45, there is repetition of a sentence. Such typographical issues should be corrected throughout the manuscript.
Reviewer 2 Report
Comments and Suggestions for Authors
This paper develops a methodology to identify false detections in the STIV results and proposes an approach to estimate missing values. Specifically, three indicators are defined to validate the STIV outcomes, and an exponential distribution of cross-section velocity is employed to estimate the missing data. The authors tested this methodology under several weather and environmental conditions.
The paper is relevant and potentially very useful for improving STIV analyses; however, some improvements are necessary before it can be considered suitable for publication.
General comments:
- The authors use the term "rectification" to refer to the process of validating results and extrapolating missing data. The use of this term should be reconsidered, as it may cause confusion with the orthorectification process, which involves transforming results from an oblique image system to a real-world reference system.
- The notation used throughout the manuscript should be thoroughly reviewed to ensure uniformity and consistency.
- The methodology includes site-specific elements, such as the selection of the cross-sectional velocity profile. Can the authors comment on this point and provide recommendations to enhance the general applicability of the proposed approach?
- The description of the statistical analysis used should be improved as it is not clear to the reader. Additionally, the term "indicators" should not be used in section 3.1, as it is already employed to refer to SNR, Symmetry, and Spectral Width.
- The proposed method was tested under three different environmental conditions (rainy, sunny, nighttime) but with varying water levels. This makes it difficult to evaluate whether the results are influenced by environmental conditions or flow conditions. Furthermore, only a single example is provided for each analysed condition.
Specific comments:
L27. In the introduction, better emphasise the advantages of using STIV compared to other image velocimetry methods.
L43-44. There is a repetition of text; please remove it.
L81. The authors explain that the search lines were set along the direction of water flow, and readers can infer that the camera is positioned perpendicular to this direction, making the search lines horizontal in the image. This should be stated more clearly in the text. Furthermore, if the camera is not installed perpendicular to the flow, what procedure do the authors suggest for identifying the search lines?
L85. When the authors say 'pixels along the search line are collected,' do they mean the intensity values of each pixel along the search line are collected?
Figure 1 needs improvement. For instance, it is not clear that panels b and c refer to a single search line. Additionally, axis labels should be included.
L104. Provide a brief description of the field calibration here.
L125. Change “the following equation” with “Equation 4”.
Figures 2 and onwards should include axis labels and/or a colormap.
L174. Discuss what it implies to use N/4 with good condition, considering that later only N/4 will be used.
Figures 6, 7, and 8 could be summarized into a single image that graphically shows the three indicators. Additionally, in Figure 7b, it is unclear which is the valley and which is the peak, as the valley appears higher than the peak. Figure 8 should be zoomed in, as the different identified points are not distinguishable.
L261. Could the threshold for spectral width vary for different study sites? If so, it should be specified and the calibration process should be outlined.
Figure 9 does not provide significant additional information and could be not included.
L273. This is site-specific (see general comments).
L277. The symbol L has already been used to indicate the length of the search line.
L283. It is not clear, please rewrite this sentence.
L291. Clarify that for each test, the velocity value along the various search lines was determined in four different ways and describe them in detail. Specifically, explain how the velocity was measured with the current meter: considering the river depth, it is likely the river is not accessible. Also, was only the surface velocity surveyed?
L304. Provide a clearer description of the equations. Is the actual value the value measured with the current meter? Is the detection value the velocity obtained using one of the other three methods? Is the mean of actual values actually used in the analysis? Does i refer to the different search lines?
L317. Please specify the model of the used camera.
Figure 11. Improve the quality of the image and the text. Additionally, the information contained is not specified in the text.
L343. What is meant by "reference point"?
L377. Do reflections and shadows affect the identification of MOS in the same way?
Test 2. To what extent are the results influenced by the environmental condition (sunny, in this case) and by the water level? To distinguish the various effects, I would suggest performing the analysis under similar environmental conditions but with different water levels, or vice versa.
Figure 14-16-18. Does panel a also show the velocity values identified as invalid using the proposed methodology? If so, please highlight them.
